# High temperature limit of photosynthetic excitons

Margus Rätsep [1], Renata Muru[1] & Arvi Freiberg [1,2]

Excitons in light-harvesting complexes are known to significantly improve solar-energy harnessing. Here we demonstrate photosynthetic excitons at super-physiological temperatures reaching 60–80 °C in different species of mesophilic photosynthetic bacteria. It is shown that the survival of light-harvesting excitons in the peripheral LH2 antennae is restricted by thermal decomposition of the pigment–protein complex rather than by any intrinsic property of excitons. The regular spatial organization of the bacteriochlorophyll *a* pigments supporting excitons in this complex is lost upon the temperature-induced breakdown of its tertiary structure. Secondary structures of the complexes survive even higher temperatures. The discovered pivotal role of the protein scaffold in the stabilization of excitons comprises an important aspect of structure–function relationship in biology. These results also intimately entangle the fundamental issues of quantum mechanical concepts in biology and in the folding of proteins.

---

[1] Institute of Physics, University of Tartu, W. Ostwald Str. 1, 50411 Tartu, Estonia. [2] Institute of Molecular and Cell Biology, University of Tartu, Riia 23, 51010 Tartu, Estonia. Correspondence and requests for materials should be addressed to A.F. (email: arvi.freiberg@ut.ee)

Excitons as delocalized collective electronic excitations of matter were first defined for regular crystalline materials[1]. The concept has evolved much over time and is now widely used for characterizing light-mediated processes in complex nanostructured materials of artificial as well as native origin, including photosynthetic systems[2]. In photosynthesis—a process that converts solar photons into energy-rich chemicals—excitons of light-harvesting complexes are considered to significantly improve the efficiency of solar-energy harnessing[3–5].

By virtue of fundamental dephasing effects that enhance with temperature, excitonic properties are expected to be highly sensitive on temperature. This aspect with respect to photosynthetic excitons has only recently gained systematic attention[6–9]. It was first experimentally proved using polarized fluorescence excitation spectroscopy that the basic exciton model[4] appropriately complemented by electron–phonon coupling and static disorder was perfectly adequate for description of photo-excitations in the photosynthetic antenna complexes all the way from cryogenic temperatures to close ambient temperatures[7–9]. A relatively minor (13%) narrowing of the exciton-state manifold (exciton band), and thus the exciton coupling energy, was observed between 4.5 and 263 K[8]. The phenomenal thermal robustness of photosynthetic antenna excitons was later confirmed by sophisticated ab initio computational modeling, where the protein and external environment effects were taken into account in terms of polarizable medium[6]. The temperature-dependent evolution of excitons in light-harvesting organelles of increasing structural complexity, from detergent-isolated complexes to complete bacterial cells, was further followed[9].

In the present work, we set to investigate the high-temperature limit for the existence of photosynthetic excitons, as well as the physical factors that determine this limit. To achieve this, we study in parallel the circular dichroism (CD) and absorption spectra of the LH2 peripheral antenna complexes upon varying the temperature above the ambient temperature. Two different mesophilic purple photosynthetic bacteria *Rhodoblastus*

*acidophilus* and *Rhodobacter sphaeroides* are explored in order to establish generality of the observations. The favored habitat temperature for these bacteria is around 30 °C (304 K). Simultaneous monitoring at different temperatures of the CD and absorption spectra over a wide spectral range spanning from far-ultraviolet to near-infrared allows distinguishing separate responses of the protein and pigment components of the complex, required to answer the raised questions. The results provide decisive evidence that excitons are indeed one of the key functional elements of photosynthetic organisms.

## Results

**Optical spectra of the LH2 complex at ambient temperature.** The LH2 complex from *R. acidophilus* accommodates 27 bacteriochlorophyll *a* (BChl *a*) molecules organized into two concentric rings called B800 (9 BChl *a* molecules arranged in the cytoplasmic (outer) side of the membrane protein) and B850 (18 BChl *a* molecules in the periplasmic (inner) side), see Fig. 1a[10]. The nano-sized pigment rings (with 6.22 and 5.30 nm diameter in B800 and B850, respectively) are non-covalently squeezed between the membrane's α-helical α- and β-polypeptides. An additional nine carotenoid pigment molecules (rhodopin glucoside, one per basic building block of α- and β-polypeptide) interconnect the B850 and B800 rings. The B800 and B850 pigment arrangements give rise to intense absorption bands of the LH2 complex in near-infrared, around 800 and 850 nm (Fig. 1b). The exciton nature of these bands, which is related to the $Q_y$ singlet electronic transition of the BChl *a* pigments, is generally recognized[4,11–15].

Figure 1b displays the overview of absorption and CD spectra of the LH2 complex from *R. acidophilus* at room temperature. The CD spectroscopy, providing information on the chirality of individual molecules and the interactions between guest and host molecules in condensed matter, is a powerful method for exploring the excitonically coupled photosynthetic antenna assemblies[16,17]. The spectra can be divided into protein and

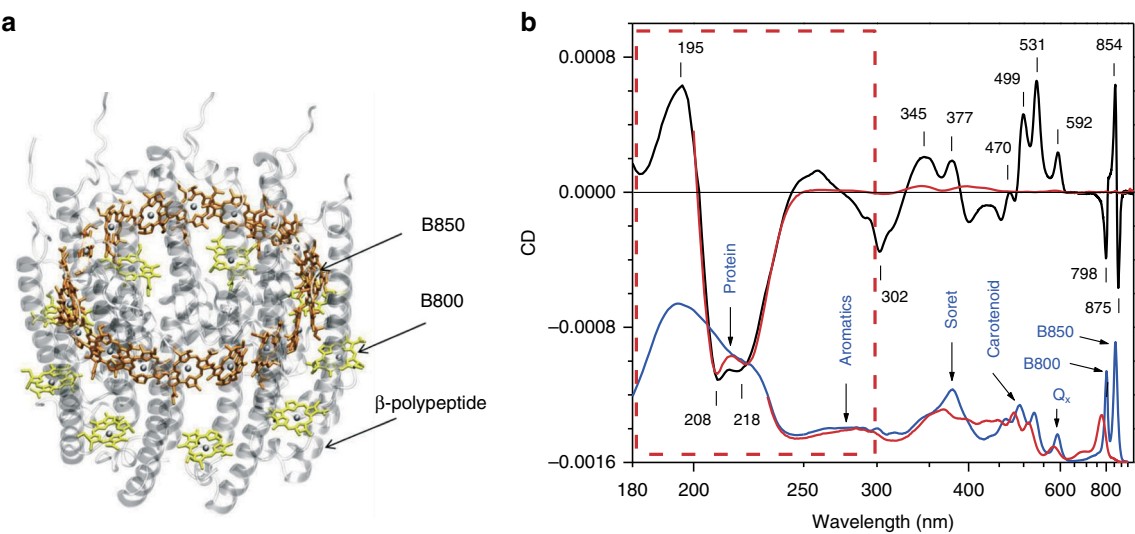

**Fig. 1** Structure and optical spectra of the LH2 complex from *Rhodoblastus acidophilus*. **a** Simplified structure of the LH2 complex emphasizing cyclic arrangement of the BChl pigment chromophores and α–helical secondary-structure conformation of transmembrane polypeptides. For clarity, carotenoid molecules are left out and only half of the polypeptides (β-) are presented. © aegon/Wikimedia Commons/GFDL and CC-BY-SA, with modifications. **b** CD (black line) and absorption (blue line) spectra of the isolated LH2 complex at 20 °C. Reciprocal (linear in energy) wavelength scale is used. Numbers at corresponding vertical lines indicate the positions of the CD spectral lines in nanometers. The area surrounded by the dashed red line delineates the protein range. The absorption spectrum with optical density of 0.58 at 858 nm was multiplied by a factor of $1.2 \times 10^{-3}$ and shifted vertically to show absorption and CD spectra in one figure. Shown by red line are the corresponding CD and absorption spectra measured at 20 °C after heating up the sample to 70 °C and cooling it down

pigment parts at ~300 nm, as shown in Fig. 1b. Collective excitations of secondary-structure peptide bond (195–220 nm) and aromatic amino acid (260–295 nm) chromophores are responsible for the significant absorbance in the protein range (<300 nm). In the pigment range (>300 nm) the peaks of absorption spectrum belong to different electron-vibrational transitions either in BChl (377, 592, 802, and 858 nm) or in carotenoid (470–530 nm) molecules. The CD spectrum in the pigment range reveals positive–negative differential structure for most of the pigment absorption bands, which is typical for excitons[16]. A distorted shape of the CD spectrum of B800 excitons with a lone negative band at 798 nm can be explained by an overlap with the strong positive high-energy exciton component of the B850 system. The protein-range CD spectrum exhibits the well-known features (two negative peaks at 218 and 208 nm and a positive peak at 195 nm) of the peptide bond exciton of transmembrane polypeptides in α-helical conformation[18]. The CD spectrum of aromatic residues is very weak and is also interfered by a strong negative signal of pigment origin.

**Spectral changes observed at elevated temperatures**. To gain knowledge about the relationship between the excitonic structure of pigment and the protein thermal stability, the spectra at different temperatures between 20 and 80.5 °C were recorded. The CD spectra as presented in Fig. 2a show relatively minor changes of excitonic structure until 50 °C in both pigment and protein ranges. Above this temperature, the CD signal recorded in the pigment range rapidly dropped, totally vanishing at around 60 °C. This is in stark contrast with the protein spectra, which at similar temperatures demonstrated but minor changes, as shown in Fig. 2b. Even at 80.5 °C roughly half of the protein's CD signal persisted.

Focusing on the near-infrared range of BChl *a* absorption, the spectral changes with increasing temperature were likewise marginal within the physiologically competent range of temperatures, see Fig. 3a and also Supplementary Fig. 1. Beginning from about 40–50 °C, however, a concerted decrease of the B850 and B800 bands occurred. Simultaneously, a new band around 775 nm appeared that finally dominated the whole spectrum. By spectral position and shape, this thermo-product band can be assigned to individual solubilized BChl *a* molecules, irreversibly detached from the binding pockets of LH2 due to denaturation of the protein. The weak shoulders observable at 70 °C are likely associated with the forms of LH2 where the BChl *a* pigments are either partially (peak around 845 nm) or fully oxidized (695 nm)[19,20]. It is important to notice that at all temperatures the shape of the residual absorption spectrum (shown in red in the inset of Fig. 3a) obtained upon subtraction of the described denaturation components is very similar to the original spectrum of intact LH2 complexes. Along with the selective and single-molecule spectroscopy data, which have revealed heterogeneity of various spectroscopic properties of the antenna complexes[21,22], these observations imply variant resistance against the temperatures of different LH2 complexes in the sample ensemble. Similar heterogeneity with respect to pressure, another major thermodynamic parameter, has also been observed[23].

Shown in Fig. 3b are the CD spectra in the same pigment spectral range. To compare the spectral shapes recorded at different temperatures, the spectra were normalized to the B850 absorption band intensity at specific temperatures. As seen, the spectral bands retain their shape almost until the total loss of spectral intensity taking place at ~62.5 °C, see below. The only significant change in the CD spectra is a ~2-nm blue shift, consistent with the shift observed in the absorption spectrum. CD spectra from individual, solubilized BChl *a* molecules are at least

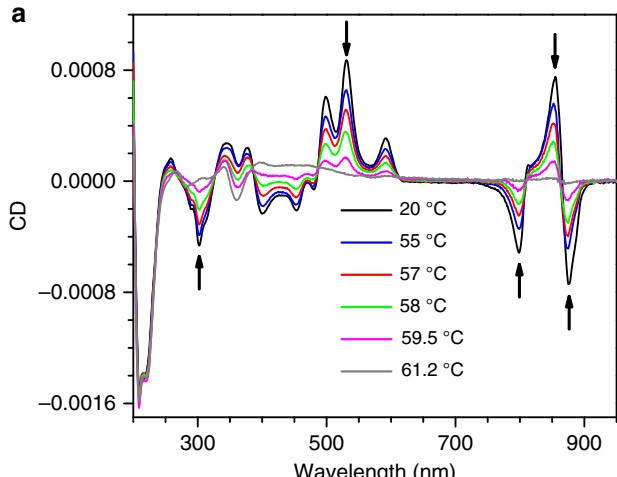

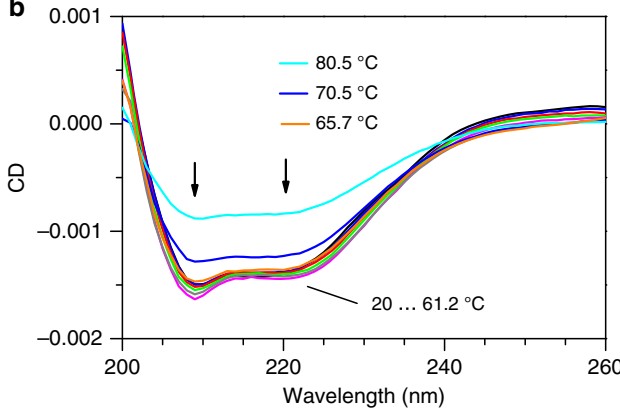

**Fig. 2** Temperature dependence of CD spectra. **a** CD spectra of the LH2 complex from *R. acidophilus* at different temperatures. The initial optical density of the sample in 1 mm cuvette at 858 nm was 0.58. **b** The zoomed-in protein region of the CD spectra. Arrows indicate the lines selected for presentation in Fig. 4

an order of magnitude weaker than from B800/B850 excitons[24]. Therefore, in contrast to the absorption spectrum, there is no apparent CD activity around 775 nm when temperature increases. Figure 3a, b shows near invariance of the shapes of the remaining absorption and CD spectra up until thermal rupture of the protein structure which suggests the protein scaffold has a pivotal role in the survival of photosynthetic excitons.

The wide-range absorption and CD spectra of the thermal degradation product recorded at 20 °C are shown in Fig. 1b with a red line. In the pigment area, significant modifications of the absorption spectrum and almost complete loss of CD spectral structure was observed, in contrast to only little changes in the protein area. Following is a more detailed study of this phenomenon. Figure 4 shows the temperature dependence of major CD peaks corresponding to pigment and protein excitons. While the traces related to the peptide bond exciton of transmembrane α-helices (two lowest curves in Fig. 4) show only a limited decrease all the way to the boiling water temperature, the peaks of pigment excitons follow a more intricate path. The peak intensities, nearly constant up to 30 °C, start a simultaneous progressive decrease until vanishing at a similar temperature around 62 °C. The change of the protein CD intensities, as measured at 218/208 nm, at this critical temperature is present, but very small (see the rectangular inset in Fig. 4).

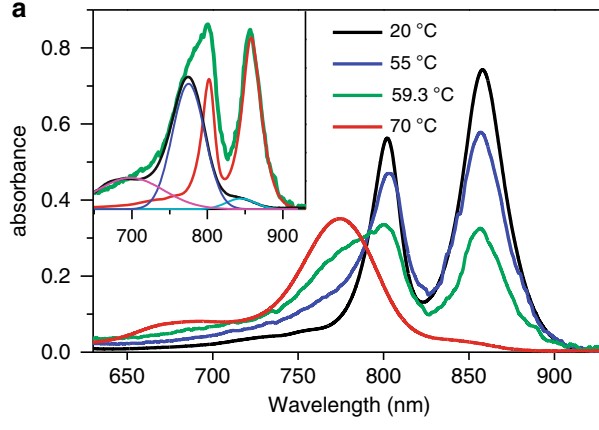

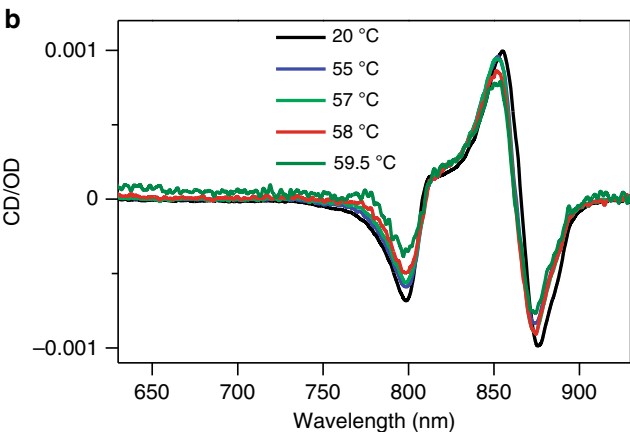

**Fig. 3** Temperature dependence of absorption and normalized CD spectra. **a** Absorption spectra of LH2 complexes at various temperatures indicated. The inset shows a deconvolution of the absorption spectrum measured at 59.3 °C (green line) into intact LH2 (red line) and denatured temperature-induced product (black line) components. The shape of the denaturized component is further deconvoluted into three Gaussian curves corresponding to different forms of absorbing species: monomeric BChls (blue line), oxidized (magenta line), and partially disintegrated (cyan line) LH2 complexes. **b** CD spectra normalized to the B850 absorbance of LH2 complexes at the corresponding temperatures indicated

This suggests that the parts of the protein scaffold that support pigment and protein excitons not only have rather different thermal stability, but must also be functionally isolated from each other. Although the spectra simultaneously change with time, especially at higher temperatures (see Supplementary Fig. 2), the change rate is slow compared to the experimental timeframe and may qualitatively be considered irrelevant.

Similar measurements were performed on LH2 complexes from another mesophilic photosynthetic bacterium *Rba. sphaeroide*s, with the result that pigment excitons in this species tolerate even greater temperatures (reaching 81 °C), see Supplementary Figs 3 and 4 in Supplementary Information. To ascertain that purification of the membrane proteins into detergent micelles is not the main cause of the above effects, detergent-isolated and native membrane-bound LH2 complexes were studied in parallel. Inspection of Supplementary Figs 3 and 4 for isolated complexes and Supplementary Figs. 5 and 6 for membrane complexes demonstrates the generality of the main observations. Also, in all cases, thermal degradation of excitonic structures was irreversible once the sample temperature was raised above the critical temperature—either ~62 °C (*R. acidophilus*) or ~81 °C (*Rba. sphaeroide*s).

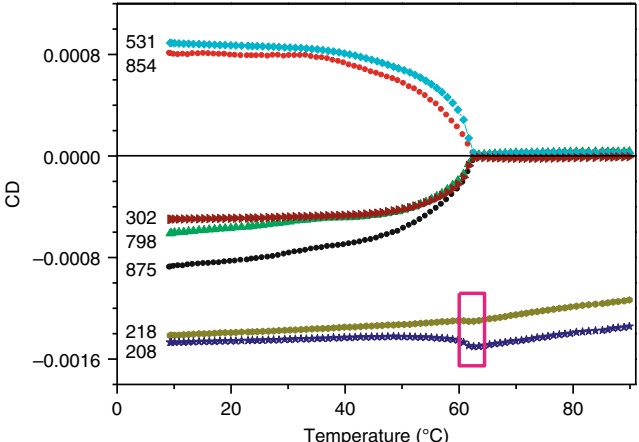

**Fig. 4** Intensity of selected CD lines as a function of temperature. Temperature dependence of the CD signal intensities at the wavelengths indicated in nanometers. The initial optical density of the sample in 1 mm cuvette at 858 nm was 0.62. The sample temperature was raised with a rate of 1 °C/min; measurement time per temperature point was ~1 min. The pink rectangle highlights the change of the protein CD intensities

## Discussion

We have thus discovered extreme robustness against temperature of excitons in mesophilic photosynthetic bacteria. The pigment excitons cease to exist as a result of thermal disintegration of the pigment binding pockets of the protein scaffold and not because of some internal property of excitons like diminishing ratio of the exciton coupling energy to the reorganization energy with increase of temperature. This is surprising because strong electron–phonon coupling was established for the B850 excitons[21], causing their self-trapping within less than 200 fs[25]. Thermal stability of the protein-related excitons is higher still; they survive at least 90 °C, the limiting temperature in current experiments.

To reconcile these apparently contradictory observations, one has to keep in mind a special hierarchical (primary, secondary, tertiary etc.) build-up of protein structures as well as a related order of the forces responsible for different levels of protein folding. Here the pigment and protein excitons are evidently supported by different hierarchies of the LH2 protein organization—pigment excitons by tertiary structure and protein excitons by secondary structure. Thermal energy required for destroying the specific tertiary fold of the protein responsible for keeping the correct pigments formation for exitonic effects to arise can be estimated as $k_B T_c$, where $T_c$ is the above critical temperature in K units. Its value (2.79 kJ/mol for *R. acidophilus* or ~ 3.00 kJ/mol for *Rba. sphaeroide*s) is only marginally greater than the average thermal energy of the habitats of these bacteria. Also, in ref. [26] it was found that the binding energy between the LH2 protein subunits (i.e., between its tertiary structure elements) is in the order of $k_B T$ at ambient temperature (~ 2.44 kJ/mol). In contrast, secondary-structure elements such as membranes spanning α-helices are known to be highly resistant to thermal denaturation. The free energy of stabilization of a transmembrane helix has been estimated to be an order of 290 kJ/mol[27]. Therefore, it is only feasible that the higher-order fold governed by relatively weaker forces unfolds at lower temperatures compared to the lower-order and more stable fold.

In conclusion, this work explored the properties of photosynthetic excitons as a function of temperature across the ambient and super-physiological temperatures with the aim of finding out the limiting factors for the existence of these biological quantum photo-excitations. Our experimental results and analysis revealed that the pigment excitons exist in the pigment–protein complexes

of photosynthetic purple bacteria until the tertiary structure of the supporting protein scaffold breaks down. The results confirm the validity of using quantum mechanical approaches to describe biological processes, simultaneously outlining important aspects of the structure–function relationship in biology.

## Methods

**Samples**. The concentrated LH2 complexes from *R. acidophilus* and *Rba. sphaeroides* were prepared as described earlier[28] and stored at −78 °C in deep freezer. Before use, the samples were diluted with 20 mM TRIS pH 8.0 (20 mM Hepes pH 7.5 for *Rba. sphaeroides*) buffer to obtain a reasonable optical density. To keep the complexes isolated a specific detergent 0.1% of LDAO (1.0% of β-OG) was added to the final sample solution. No detergent was used for the membrane LH2 complex.

**Spectroscopy**. The CD and absorption spectra were measured using a Chirascan Plus spectrophotometer (Applied Photophysics, UK). Some absorption spectra were also recorded with a 0.3-m spectrograph Shamrock SR-303i equipped with a thermo-electrically cooled CCD camera DV420A-OE (both Andor Technology, UK). As a light source for absorption measurements, a high-stability tungsten lamp BPS100 (BWTek, USA) was used. Quartz cuvettes (Hellma Analytics, Germany) with 0.1–1.0 mm path length were used at 10–90 °C in a thermostabilized cell holder.

**Data availability**. The data that support the findings of this study are available from the corresponding author on request.

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

## Acknowledgements

This work was supported by the Estonian Research Council (grant IUT02-28) and the Australian Research Council Discovery Project (grant DP150103137). We thank Professors R. Cogdell (University of Glasgow) and N. Hunter (University of Sheffield) for kindly donating the samples.

## Author contributions

A.F. designed the project. R.M. and M.R. performed the experiments. A.F. and M.R. wrote the manuscript.

## Additional information

**Competing interests:** The authors declare no competing financial interests.

