## [Peer review file · Nature Communications]

Reviewers' comments:

Reviewer #1 (Remarks to the Author):

The paper by Ratsep et al. presents a spectroscopic investigation on the "robustness" of the excitonic nature of LH2 going from room to super-physiological temperatures.

The study compares absorption and CD spectra at different temperatures of two LH2 complexes and it analyzes the spectral changes both in the protein and the pigment regions.

The obtained results are very interesting as they clearly show the extreme robustness against temperature for pigment excitons in the two LH2 antenna complexes here investigated. This is indeed an important observation which supports the idea that Nature has used excitons to make the light harvesting function more "robust" with respect to external perturbations.

The data are presented in a clear way and the paper is well written. There is however an aspect that should be improved.

The authors do not give details about the preparation of the complexes, but they refer to an old paper by Cogdell et al. (1983). I guess that during this procedure, the "environment" of LH2 changes significantly with respect to the natural one.

Can this change affect the response of the system to temperature?

The authors should give some comments about this issue.

A minor issue: in the Intro the authors refer to quantum mechanical studies but the cited references (3-5) do not report any calculation of this type. Maybe the authors wanted to refer to Ref. 6.

Reviewer #2 (Remarks to the Author):

The paper describes a set of experiments on LH2s from photosynthetic bacteria where the excitonic structure of the bacteriochlorophylls and the protein secondary structure is followed as a function of temperature. It is concluded that the excitonic structure of the pigments is 'denatured' at much lower temperatures than the protein secondary structure, which is a nice finding but maybe not so surprising. What is surprising is that the authors see both the absorption and CD of the pigments disappear simultaneously. In LH2 the excitonic absorption depends mainly on nearest neighbor interactions while the CD is sensitive to interactions between pigments a quarter of the ring away (see several papers by Koolhaas et al on the CD of LH2). Naively one would have thought that when the rings fall apart at elevated temperatures, the CD would be much more sensitive than the linear absorption. Do these authors then believe that their results show that the rings fall apart cooperatively? That there is no intermediate with still a redshifted absorption due to nearest neighbor interactions but a strongly diminished CD. Could the authors measure the polarized fluorescence as a function of temperature to maybe establish this phenomenon?

I could recommend publication of these results if these kind of experiments plus discussion are added.

Reviewer #3 (Remarks to the Author):

In "High-temperature limit of photosynthetic excitons", the authors present temperature-dependent linear absorption and CD spectra to explore the loss of excitonic and molecular structure at high temperatures in the photosynthetic antenna protein LH2 from purple bacteria.

LH2 consists of 8-10 subunits (depending on species) arranged to form a cylindrical structure, with pigments held between subunits within the protein scaffold. They observe that the excitonic structure is lost upon breakdown of the tertiary structure, before the loss of the secondary structure of the protein. Furthermore, the data shows that the excitonic structure, as measured by CD, remains the same until the breakdown of the tertiary structure.

The temperature dependent CD is interesting, and so this work is suitable for publication. However, the loss of excitonic structure with the loss of tertiary structure is unsurprising, as discussed in more detail below. As a result, I believe the work is appropriate for a field-specific journal, and not of sufficient general interest to warrant publication in Nature Communications.

1. The tertiary structure is responsible for holding the pigments in fixed positions and relative orientations, which controls the coupling and thus the excitonic structure. The authors should clarify why the dependence of the excitonic structure on tertiary structure is surprising. Although the authors refer to dephasing, the linewidth of LH2 is dominated by static disorder, so a temperature-induced change in the dephasing rate would have minimal effect on the linear spectra presented here.
2. The observation of loss of tertiary structure well before denaturation of the subunits is consistent with previous measurements of molecular forces in LH2. The binding between subunits was found to be $\sim kT$ and the folding energy was found to be $\sim 90 kT$ (doi: 10.1073/pnas.1004205108). Given that their results are consistent with this previous work, the authors may want to refer to this work.
3. The B800 pigments are more weakly bound than the B850 pigments, and thus often fall out of the protein structure first. It would be interesting to further quantify the data in Fig. 3b to see whether there is any difference between the temperature-dependent pigment occupancy of the B800 and B850 bands.
4. On page 6 the authors state "These observations imply variant resistance against temperature of different LH2 complexes". However, if the loss of tertiary structure is a stochastic process, some complexes might remain intact without any variation, i.e., a Boltzmann distribution of intact and denatured complexes based on the temperature and energy required for protein destruction. The authors should clarify how they ruled out this possibility.

Response (in *bold*) to the Reviewers' comments

Reviewer #1:

The obtained results are very interesting as they clearly show the extreme robustness against temperature for pigment excitons in the two LH2 antenna complexes here investigated.

This is indeed an important observation which supports the idea that Nature has used excitons to make the light harvesting function more “robust” with respect to external perturbations.

The data are presented in a clear way and the paper is well written. There is however an aspect that should be improved.

The authors do not give details about the preparation of the complexes, but they refer to an old paper by Cogdell et al. (1983).

To keep the number of references manageable, we have generally followed the policy to cite original works. That’s why we referred to an old paper of Cogdell. From another side, the sample used in this work comes from the Cogdell’s lab, the most authoritative one concerning preparation and crystallisation of this specific sample. To the best of our knowledge, the procedure of preparation hasn’t changed ever since the method was originally developed. Other labs in the field have basically copied this procedure, although different buffers and detergent for different species have been used.

I guess that during this procedure, the “environment” of LH2 changes significantly with respect to the natural one. Can this change affect the response of the system to temperature? The authors should give some comments about this issue.

We have specifically studied this question and the answer is no, please see Supplementary Information where the data for detergent-isolated and native membrane embedded LH2 complexes of *Rba. sphaeroides* are presented. We have stressed this aspect more clearly in the revised text, line 169 and further.

A minor issue: in the Intro the authors refer to quantum mechanical studies but the cited references (3-5) do not report any calculation of this type. Maybe the authors wanted to refer to Ref. 6.

Indeed. Must be a formatting error; corrected in the revised text, line 37.

Reviewer #2:

It is concluded that the excitonic structure of the pigments is 'denatured' at much lower temperatures than the protein secondary structure, which is a nice finding but maybe not so surprising. What is surprising is that the authors see both the absorption and CD of the pigments disappear simultaneously. In LH2 the excitonic absorption depends mainly on nearest neighbor interactions while the CD is sensitive to interactions between pigments a quarter of the ring away (see several papers by Koolhaas et al on the CD of LH2). Naively one would have thought that when the rings fall apart at elevated temperatures, the CD would be much more sensitive than the linear absorption. Do these authors then believe that their results show that the rings fall apart cooperatively? That there is no intermediate with still a redshifted absorption due to nearest neighbor interactions but a strongly diminished CD. Could the authors measure the polarized fluorescence as a function of temperature to maybe establish this phenomenon?

Generally speaking, the correlated decay of absorbance and CD shouldn't be that surprising. CD is only observable if the sample absorbs light. In a more detailed level, however, temperature dependences of absorption and CD are not quite identical in our samples. As shown in Fig. 4 (as well as Supplementary Figs. 2, 4, and in a newly designed Supplementary Fig. 6), certain variations are observed even between temperature dependences of different CD peaks. Spectral intermediates can be easily created in LH1 complexes either by detergent treatment or under high hydrostatic pressure. To the best of our knowledge, no such phenomena have been ever observed for LH2 complexes. The more robust structure of LH2 has been explained by existing differences between the pigment-protein hydrogen bonding in LH2 and LH1 complexes. We also do not quite agree that excitonic absorption in the LH2 complex depends mainly on nearest neighbour interactions. This is too simplistic, if to acknowledge the fact that in the B850 oligomer the intra-dimer interactions are as strong (or even stronger) as the inter-dimer interactions. Complex relationships between the absorption and CD spectra of LH2 complexes as a function of temperature were recently theoretically thoroughly analysed (see ref. 6).

Reviewer #3:

The temperature dependent CD is interesting, and so this work is suitable for publication. However, the loss of excitonic structure with the loss of tertiary structure is unsurprising, as discussed in more detail below. As a result, I believe the work is appropriate for a field-specific journal, and not of sufficient general interest to warrant publication in Nature Communications.

Establishing a correlation between the loss of excitonic structure and the protein tertiary structure wasn't a goal of the current study, but part of a series of explanations of our experimental observations.

The main task of this work was to determine the boundary temperature associable with functional photosynthetic excitons and to understand the factors limiting this temperature. The core of the problem is physical, while its application is biological. The molecular (Frenkel) exciton as a physical concept depends on precise phase relationships

between the electronic wavefunctions of nearby molecules. Therefore, they are bound to be sensitive to temperature that distorts relative phases of the wavefunctions. A common understanding is that excitons cease to exist (become localized) when dephasing develops very fast. In molecular crystals this happens way below room temperature. Yet photosynthetic pigment excitons function perfectly well at ambient temperatures. The question is why?

Here, we firstly demonstrated that pigment excitons even in mesophilic photosynthetic bacteria survive up to 80°C, well beyond their habitat temperature. This was a surprise. Then we established that the very high survival temperature of the excitons is not limited by thermally-induced dephasing of excitons (a physical or internal reason) but by the mechanical strength of the protein surroundings (biological or external reason). This was second surprise. Taken together, these results are at least for physical biologist both original and surprising.

1. The tertiary structure is responsible for holding the pigments in fixed positions and relative orientations, which controls the coupling and thus the excitonic structure. The authors should clarify why the dependence of the excitonic structure on tertiary structure is surprising.

See above general explanation/justification.

Although the authors refer to dephasing, the linewidth of LH2 is dominated by static disorder, so a temperature-induced change in the dephasing rate would have minimal effect on the linear spectra presented here.

This is a total misunderstanding of experimentally well studied and understood question. Many subtleties aside such as what to call static and what dynamic disorder in specific experimental situations, static disorder in the absorption spectrum of LH2 complexes dominates only at cryogenic temperatures, below about 100 K (see, e.g. doi:10.1016/j.bbabi.2011.11.019 and refs. therein). At room temperature and above the LH2 lineshapes are generally homogeneously broadened.

2. The observation of loss of tertiary structure well before denaturation of the subunits is consistent with previous measurements of molecular forces in LH2. The binding between subunits was found to be $\sim kT$ and the folding energy was found to be $\sim 90 kT$ (doi: 10.1073/pnas.1004205108). Given that their results are consistent with this previous work, the authors may want to refer to this work.

We thank the Reviewer for this useful reference now included into the reference list and adequately commented (lines 194-195 in the revised text).

3. The B800 pigments are more weakly bound than the B850 pigments, and thus often fall out of the protein structure first. It would be interesting to further quantify the data in Fig. 3b to see whether there is any difference between the temperature-dependent pigment occupancy of the B800 and B850 bands.

This effect is generally species dependent. For purified complexes, it also depends on specific procedures of isolation and chemicals used. In current experiments, we didn't observe any significant difference in the decay of B800 and B850 bands/pigments, see new Supplementary Figure 6 in Supplementary information.

4. On page 6 the authors state "These observations imply variant resistance against temperature of different LH2 complexes". However, if the loss of tertiary structure is a stochastic process, some complexes might remain intact without any variation, i.e., a Boltzmann distribution of intact and denatured complexes based on the temperature and energy required for protein destruction. The authors should clarify how they ruled out this possibility.

Heterogeneity is an intrinsic property of various protein ensembles, well demonstrated by different single-molecule experiments, including our own. In every distribution and at all temperatures there are proteins susceptible for denaturation and those which are very stable. The Boltzmann distribution reveals itself in the fact that we observe growing amount of irreversibly destroyed proteins with increasing temperature. There is thus no contradiction and no ruling out possibilities.

REVIEWERS' COMMENTS:

Reviewer #1 (Remarks to the Author):

All the points raised in the previous round of review have been satisfactorily addressed by the authors in the revised manuscript.

I recommend publication of these results without any further change.

Reviewer #2 (Remarks to the Author):

I have no further comments on the manuscript, it can be published as it is.

Reviewer #3 (Remarks to the Author):

The authors assert that the lack of thermally-induced dephasing of the Frenkel excitons is a surprising result. In general, dephasing (or, more precisely, localization) of the wavefunction depends on the relative amplitude of the coupling to the environment and the coupling between the monomers that gives rise to the delocalized wavefunction (see, e.g., *Photosynthetic Excitons* by van Amerongen, et al.). The authors assert that this is surprising due to the difference in dephasing between molecular crystals and photosynthetic LHCs. However, the coupling to the environment is quite different in molecular crystals. For example, molecular crystals have several modes with much stronger Huang-Rhys factors than LHCs. PTCDA has two modes with Huang Rhys factors of ~ 0.4 (DOI: 10.1103/PhysRevB.80.115309) and, as some of the authors have shown, the strongest mode of BChl a has a Huang-Rhys factor of 0.028 (DOI: 10.1063/1.3518685). Given this difference, if the lack of dephasing being surprising is the central claim of the manuscript, the surprising aspect needs to be better justified by an explicit experimental or theoretical result such as one showing that the coupling to the protein is much stronger than the coupling between pigments, yet dephasing does not occur. Without this type of explicit justification, the basis for the claim that the results are surprising is not clear.

Secondly, the authors introduce the term "variant resistance" for describing the loss of tertiary structure. Observing different behaviors in different molecules is due to, in general, two types of effects: (1) it is observation of a stochastic process where denaturation occurs at different times for different proteins due to the stochastic nature of the process. If the same protein were to undergo a denaturation several times, it would occur at a slightly different time each cycle due to stochasticity. Thus, repeated measurements of the denaturation of either one protein or the ensemble would give rise to a single denaturation rate; or (2) different proteins have different structures/interactions such that denaturation occurs at different times due to intrinsic differences. If proteins of one type were to undergo a denaturation several times, it would occur on average with a different timescale than another type. Repeated measurements of an individual type of protein would give rise to a single denaturation rate, while measurements of the ensemble would give rise to the sum of both rates. The term "variant resistance" suggests (2), yet the authors agree the observations are consistent with (1). Given that there is no reason to expect perfect synchronization for a stochastic process, the use of the term is misleading and, if the authors wish to use it, they should clarify their definition.

Response (in bold) to the Reviewers' comments

Reviewer #1:

All the points raised in the previous round of review have been satisfactorily addressed by the authors in the revised manuscript.

I recommend publication of these results without any further change.

Reviewer #2:

I have no further comments on the manuscript, it can be published as it is.

Reviewer #3:

The authors assert that the lack of thermally-induced dephasing of the Frenkel excitons is a surprising result. In general, dephasing (or, more precisely, localization) of the wavefunction depends on the relative amplitude of the coupling to the environment and the coupling between the monomers that gives rise to the delocalized wavefunction (see, e.g., Photosynthetic Excitons by van Amerongen, et al.). The authors assert that this is surprising due to the difference in dephasing between molecular crystals and photosynthetic LHCs. However, the coupling to the environment is quite different in molecular crystals. For example, molecular crystals have several modes with much stronger Huang-Rhys factors than LHCs. PTCDa has two modes with Huang Rhys factors of ~ 0.4 (DOI: 10.1103/PhysRevB.80.115309) and, as some of the authors have shown, the strongest mode of BChl a has a Huang-Rhys factor of 0.028 (DOI: 10.1063/1.3518685). Given this difference, if the lack of dephasing being surprising is the central claim of the manuscript, the surprising aspect needs to be better justified by an explicit experimental or theoretical result such as one showing that the coupling to the protein is much stronger than the coupling between pigments, yet dephasing does not occur. Without this type of explicit justification, the basis for the claim that the results are surprising is not clear.

Author reply:

We acknowledge this account, which to our mind only emphasises the generality of the problem raised. There indeed are significant differences between the well-studied excitons in molecular crystals and the so called photosynthetic excitons worth further elaboration. We would like to further detail that

- (i) The referred to by the Reviewer electron-intramolecular vibrational interaction appears to play insignificant (and understood) role in the dephasing of excitons in the LH2 complex, see Ref. 21 in the revised manuscript.**
- (ii) The more relevant to this situation electron-phonon coupling is, in contrast, strong with the apparent Huang-Rhys factor measured at low temperatures >2 (see Refs 7, 8, 14, 21 in the revised manuscript). Therefore, we remain to be intrigued why this doesn't cause abrupt localization of the photosynthetic**

excitons at physiological temperatures, hoping to return to this question in our subsequent studies.

- (iii) **The energetic contributions of exciton-phonon coupling (its reorganization energy) and static disorder, which both, though via different mechanisms, contribute to the localization of exciton wavefunctions are of similar order of magnitude as the resonant (exciton) coupling energy between the nearest-neighbour pigments in the LH2 complex. This situation of missing small/large parameters is theoretically most demanding, solvable only by brute-force numerical calculations.**

Secondly, the authors introduce the term “variant resistance” for describing the loss of tertiary structure. Observing different behaviors in different molecules is due to, in general, two types of effects: (1) it is observation of a stochastic process where denaturation occurs at different times for different proteins due to the stochastic nature of the process. If the same protein were to undergo a denaturation several times, it would occur at a slightly different time each cycle due to stochasticity. Thus, repeated measurements of the denaturation of either one protein or the ensemble would give rise to a single denaturation rate; or (2) different proteins have different structures/interactions such that denaturation occurs at different times due to intrinsic differences. If proteins of one type were to undergo a denaturation several times, it would occur on average with a different timescale than another type. Repeated measurements of an individual type of protein would give rise to a single denaturation rate, while measurements of the ensemble would give rise to the sum of both rates. The term “variant resistance” suggests (2), yet the authors agree the observations are consistent with (1). Given that there is no reason to expect perfect synchronization for a stochastic process, the use of the term is misleading and, if the authors wish to use it, they should clarify their definition.

Author reply:

There must be a misunderstanding. By “variant resistance” we not only suggest the presence of type (2) effects (Reviewer’s notation) but also agree with it. This strong understanding of ours is based on a long time experience with the low-temperature selective spectroscopy and single-molecule spectroscopy of photosynthetic complexes revealing heterogeneous ensembles, i.e., where the ensemble members have clearly variant properties, justifying the use of “variant resistance”.

Respective sentence in p. 5 (lines 101-104) of the revised text was modified (plus two relevant references 21, 22 added) to clarify this aspect.